# Synthesis and Antimicrobial Activity of Short Analogues of the Marine Antimicrobial Peptide Turgencin A: Effects of SAR Optimizations, Cys-Cys Cyclization and Lipopeptide Modifications

**DOI:** 10.3390/ijms232213844

**Published:** 2022-11-10

**Authors:** Hymonti Dey, Danijela Simonovic, Ingrid Norberg-Schulz Hagen, Terje Vasskog, Elizabeth G. Aarag Fredheim, Hans-Matti Blencke, Trude Anderssen, Morten B. Strøm, Tor Haug

**Affiliations:** 1The Norwegian College of Fishery Science, Faculty of Biosciences, Fisheries and Economics, UiT The Arctic University of Norway, NO-9037 Tromsø, Norway; 2Department of Pharmacy, Faculty of Health Sciences, UiT The Arctic University of Norway, NO-9037 Tromsø, Norway

**Keywords:** AMPs, Cys-Cys cyclic peptides, lipopeptides, short antibacterial peptides, structure–activity relationship, mechanism of action

## Abstract

We have synthesised short analogues of the marine antimicrobial peptide Turgencin A from the colonial Arctic ascidian *Synoicum turgens.* In this study, we focused on a central, cationic 12-residue Cys-Cys loop region within the sequence. Modified (tryptophan- and arginine-enriched) linear peptides were compared with Cys-Cys cyclic derivatives, and both linear and Cys-cyclic peptides were N-terminally acylated with octanoic acid (C_8_), decanoic acid (C_10_) or dodecanoic acid (C_12_). The highest antimicrobial potency was achieved by introducing dodecanoic acid to a cyclic Turgencin A analogue with low intrinsic hydrophobicity, and by introducing octanoic acid to a cyclic analogue displaying a higher intrinsic hydrophobicity. Among all tested synthetic Turgencin A lipopeptide analogues, the most promising candidates regarding both antimicrobial and haemolytic activity were **C_12_-cTurg-1** and **C_8_-cTurg-2**. These optimized cyclic lipopeptides displayed minimum inhibitory concentrations of 4 µg/mL against *Staphylococcus aureus*, *Escherichia coli* and the fungus *Rhodothorula* sp. Mode of action studies on bacteria showed a rapid membrane disruption and bactericidal effect of the cyclic lipopeptides. Haemolytic activity against human erythrocytes was low, indicating favorable selective targeting of bacterial cells.

## 1. Introduction

Antimicrobial resistance (AMR) poses a serious threat to human health worldwide. According to a recent study, an estimated 4.95 million deaths were associated with AMR globally in 2019, out of which 1.27 million deaths were directly attributable to it [1]. Due to increasing AMR, treatment of infectious diseases has become one of the greatest challenges in modern medicine [2]. One of the efforts to mitigate this threat includes the development of new antibacterial agents, which could circumvent existing resistance mechanisms by attacking new targets (i.e., having novel mechanisms of action). Although efforts have been made in this direction, progress remains rather slow [3].

Natural products have historically played an invaluable role in drug discovery and development, and most antibiotics currently in commercial use, and those being developed, are of natural origin [4]. Gene-encoded, ribosomal synthesized antimicrobial peptides (AMPs) are widespread in nature and have been identified in various species ranging from bacteria and fungi to plants, invertebrates and vertebrates (including fish, birds and mammals) [5]. In eukaryotes, they are involved in the innate immunity as a first line of defense against infectious microorganisms. These compounds, generally small, cationic, amphipathic peptides, hold promise in the fight against AMR. Due to their non-specific mechanism of action, targeting the fundamental structure of the bacterial membrane, AMPs are thought to delay the emergence of bacterial resistance [6]. Many AMPs are shown to possess selective toxicity for microbes and a broad spectrum of antimicrobial activity, acting against both Gram-positive and Gram-negative bacteria [7]. Moreover, several studies have shown a synergistic and/or adjuvant effect of AMPs with conventional antibiotics [8,9]. Due to their favorable properties, AMPs have been successfully used as templates for the development of drug candidates with improved potency and selectivity, and several natural and synthetic peptides are currently in clinical trials [10].

The marine environment, with its vast biological diversity, is shown to be a promising source for future antibiotic discoveries, including novel AMPs [11]. We have previously isolated and characterized a 36-residue long AMP, named Turgencin A, from the Arctic marine colonial ascidian *Synoicum turgens* [12], and investigated the antimicrobial activity of its shortened linear 10-residue sequence rich in cationic residues (residues 18–27 of Turgencin A) [13] (Figure 1). In the native Turgencin A peptide, this 10-residue sequence is part of a loop region in which two cysteine residues (Cys^17^-Cys^26^) are crosslinked by a disulphide bond.

In the present study, we prepared a series of 12-residue peptides (residues 17–28) encompassing this loop region. The Lys^27^ and Leu^28^ residues belonging to the original Turgencin A sequence were also included as additional cationic and lipophilic residues, respectively. Our aim was to investigate the structure–activity relationship (SAR) of a variety of modifications to the antimicrobial activity and selectivity: (1) increasing lipophilicity (by including two tryptophan residues in the PGG core sequence), (2) Lys to Arg substitutions, (3) N-terminal acylation and (4) Cys-Cys cyclization. In doing so, we wanted to gain insight into how these modifications could be best utilized to fine-tune the properties of potential novel AMP leads, increasing both their antimicrobial activity and selectivity. All peptides, like the originally isolated Turgencin A, were C-terminally amidated in order to increase the overall positive charge. Minimal inhibitory concentrations (MIC) were determined against selected Gram-positive and Gram-negative bacterial strains and fungi. Haemolytic activity (EC_50_) was tested against human red blood cells (RBCs) and a bacterial selectivity index (SI) was calculated for each peptide. Selected peptides were investigated for their antibacterial mode of action (MoA) using luciferase and fluorescence-based assays to assess the viability and integrity of the cytoplasmic inner and outer membrane of bacterial cells.

## 2. Results and Discussion

### 2.1. Peptide Design and Synthesis

All analogues of the 12-residue loop region of Turgencin A were synthesized by Fmoc solid phase peptide synthesis (SPPS) on a fully automated microwave assisted peptide synthesizer. Standard conditions were used and coupling was completed with O-(1H-6-chlorobenzotriazole-1-yl)-1,1,3,3-tetramethyluronium hexafluorophosphate (HCTU) and *N,N*-diisopropylethylamine (DIEA) as a base. A double coupling strategy was employed to ensure efficient N-terminal acylation with octanoic (C_8_), decanoic (C_10_) and dodecanoic acid (C_12_). Prior to Cys-Cys cyclization, the synthesized peptides were purified by preparative reversed-phase high-performance liquid chromatography (RP-HPLC). Cyclization of the peptides (by disulphide formation) was successfully carried out in distilled water (pH: 6.5) at room temperature with atmospheric O_2_, or by bubbling O_2_ through the aqueous solution for one to four days. The progress of cyclization was monitored by liquid chromatography–mass spectrometry (LC–MS). The mass of the final products, obtained after lyophilization, were verified by high resolution–mass spectrometry (HR–MS) (Appendix A) and the purity (>90% for all peptides) was determined by ultra-performance liquid chromatography with UV detection (UPLC-UV) (Appendix A).

### 2.2. Structure–Activity Relationship (SAR) of Cyclic Trp- and Arg-Modified Peptides

The first series of Cys-Cys cyclic peptides (**cTurg-1**–**7**) were synthesized to investigate the effects of increasing the lipophilicity of the 12-residue loop region of Turgencin A (residues 17–28) by incorporating Trp-residues, and by replacing Lys-residues with Arg-residues (Table 1). The model peptides for these modifications were based on a previously reported series of short linear Turgencin A peptides (StAMP-peptides) demonstrating improved antimicrobial activity by introducing two additional Trp-residues [13].

The first peptide in the cTurg-series, **cTurg-1**, which contained the original Turgencin A (17–28) loop sequence was, however, inactive against all tested bacterial strains (MIC ≥ 256 µg/mL), except against the sensitive strain *Corynebacterium glutamicum* (MIC: 16 µg/mL). The Trp-modified peptides, **cTurg-2**, **cTurg-3** and **cTurg-4** were derived from **cTurg-1** by substituting amino acids present in the central PGG core of Turgencin A. The central PGG sequence of **cTurg-1** was modified as follows: WWG for **cTurg-2**, WGW for **cTurg-3** and PWW for **cTurg-4** (Table 1). Compared to **cTurg-1,** the Trp-enriched cyclic peptides showed considerable improvement in activity against all bacterial strains, except for **cTurg-4** against *Pseudomonas aeruginosa* (Table 1). The highest antibacterial activity was achieved against the Gram-positive strains *Bacillus subtilis* and *C. glutamicum* (MIC: 4–8 µg/mL), as well as improved potency against *Staphylococcus aureus* and *Staphylococcus epidermidis* (MIC: 16–64 µg/mL). However, the potency against the Gram-negative strains (*Escherichia coli* and *P. aeruginosa*), though better than that of **cTurg-1**, was low (MIC: 32–256 µg/mL). This reduced activity of AMPs is most likely due to the presence of a lipopolysaccharide (LPS) layer, which is the main constituent of the outer membrane of Gram-negative bacteria and which binds to AMPs, thereby inhibiting their effect [14].

The next modification included substitution of the Lys- with Arg-residues in **cTurg-2**, **cTurg-3** and **cTurg-4**, resulting in the Arg-modified peptides **cTurg-5** (WWG), **cTurg-6** (WGW) and **cTurg-7** (PWW). These modifications led to a considerable increase in antimicrobial activity for the arginine-modified peptides against the Gram-positive bacteria, *S. aureus* and *S. epidermidis* (MIC: 8–16 µg/mL), and also against the Gram-negative bacteria *E. coli* and *P. aeruginosa* (MIC: 8–32 µg/mL) (Table 1). The most potent peptide was **cTurg-6** (WGW) with a MIC of 4–16 µg/mL against the Gram-positive bacterial strains and a MIC of 8–16 µg/mL against the Gram-negative bacterial strains. Of note, **cTurg-3** (WGW), with the same central core as **cTurg-6,** was the most potent peptide among the Lys-containing peptides, except against *P. aeruginosa*. The lowest overall antimicrobial activity for both the Lys- and Arg-containing analogues was observed for the two peptides with a PWW central core and three adjacent tryptophan residues in their sequences (**cTurg-4** and **cTurg-7**). All peptides were non-haemolytic (EC_50_: ≥ 849 µg/mL) except for **cTurg-7**, which had an EC_50_ value of 197 µg/mL against human RBCs.

### 2.3. Structure–Activity Relationship of Linear Lipopeptides

A well-established strategy for generating peptides with increased efficacy is N-terminal conjugation with aliphatic fatty acids [15,16] (Figure 2). To investigate the effects of acylation on antimicrobial activity, we decided to synthesise both linear and cyclic lipopeptide analogues of three selected peptides. Our choice of peptides for acylation was driven by the observed potencies of the previously synthesised cyclic analogues. We chose **cTurg-1** for being a mostly inactive peptide, **cTurg-2** for being the most potent peptide against *P. aeruginosa* among the Lys-containing peptides, and finally the Arg-modified peptide **cTurg-6**, which exerted the overall highest antimicrobial activity against all strains. **cTurg-2** and **cTurg-6** were also non-haemolytic (EC_50_: >1045 µg/mL).

Acylation was done with three aliphatic fatty acids: octanoic acid (C_8_), decanoic acid (C_10_) and dodecanoic acid (C_12_), since these fatty acid were previously found to improve the antibacterial activity of various peptides [17]. Of note, similar, but two-residues shorter linear analogues of these peptides (based on Turgencin A residues 18–27) have been previously reported in literature, e.g., non-acylated peptides and those without the Cys-residues [13]. For the linear peptides **C_8_-Turg-1**, **C_10_-Turg-1** and **C_12_-Turg-1** elongation of the lipid chain from 8 to 12 carbons resulted in an overall increase in antimicrobial activity (Table 1). The most potent linear lipopeptide was the longest acylated peptide **C_12_-Turg-1** with a MIC of 4–16 µg/mL against all bacterial strains. The highest increase in antimicrobial activity following acyl chain elongation was observed against *S. aureus* with improvement in MIC from 128 to 8 µg/mL, and *P. aeruginosa* with improvement in MIC from 128 to 16 µg/mL. All three peptides **C_8_-Turg-1**, **C_10_-Turg-1** and **C_12_-Turg-1** were non-haemolytic (EC_50_: >943 µg/mL).

Regarding the more lipophilic peptide, **cTurg-2** (having a central WWG region), elongation of the N-terminal acyl chain of linear analogues had an opposite effect than that observed for the linear lipopeptides based on **cTurg-1** (PGG). In other words, for the lipopeptides **C_8_-Turg-2**, **C_10_-Turg-2** and **C_12_-Turg-2**, increasing the length of the acyl chain resulted in peptides having the same or slightly reduced antimicrobial activity. One exception in this regard was the activity of **C_10_-Turg-2** against *E. coli*, with a two-fold increased potency (MIC: 8 µg/mL) compared to **C_8_-Turg-2** and **C_12_-Turg-2** (MIC: 16 µg/mL). Thus, among the linear **Turg-2** lipopeptides, **C_8_-Turg-2** and **C_10_-Turg-2** showed the highest potency against both the Gram-positive (MIC: 4–8 µg/mL) and Gram-negative bacteria (MIC: 8–16 µg/mL). Of note, all three lipopeptides **C_8_-Turg-2**, **C_10_-Turg-2** and **C_12_-Turg-2** showed the same antimicrobial activity against *B. subtilis* and *S. epidermidis* (MIC: 8 µg/mL). Compared to **C_8_-Turg-1** and **C_10_-Turg-1** (PGG core sequence), we observed improved antimicrobial activity for **C_8_-Turg-2** and **C_10_-Turg-2** (WWG core sequence) against *S. aureus*, *E. coli* and *P. aeruginosa*. However, **C_8_-Turg-2**, **C_10_-Turg-2** and **C_12_-Turg-2** displayed increasingly higher haemolytic activity from EC_50_: 198 to 55 µg/mL.

The linear lipopeptides based on the **cTurg-6** (WGW) sequence resulted in an even more noticeable reduction in antimicrobial activity following acyl chain elongation than that observed for the **C_8_-Turg-2**, **C_10_-Turg-2** and **C_12_-Turg-2** lipopeptides. Compared to **cTurg-6**, the linear lipopeptide **C_8_-Turg-6** showed a similar antibacterial effect. Further acyl chain elongation in **C_10_-Turg-6** and **C_12_-Turg-6** resulted in a significant decrease in potency, which was especially noticeable against *S. aureus* and the Gram-negative bacteria, *E. coli* and *P. aeruginosa* (MIC: 32–128 µg/mL). Additionally, **C_8_-Turg-6**, **C_10_-Turg-6** and **C_12_-Turg-6** were all rather haemolytic (EC_50_: 21–54 µg/mL). As expected, acyl chain elongation led to increased hydrophobicity for all three lipopeptide series, as monitored by their retention time on an RP-HPLC C_18_ column (Table 1).

### 2.4. Structure–Activity Relationship of Cyclic Lipopeptides

Our final modification included peptide cyclization of the previous series of lipopeptides by sidechain disulphide formation. In general, cyclization of the acylated peptides resulted in some of the most potent peptides prepared and it had some unexpected effects on their haemolytic activity (Table 1). The antimicrobial activity of the cyclic lipopeptides **C_8_-cTurg-1**, **C_10_-cTurg-1** and **C_12_-cTurg-1** was improved by increasing the acyl chain length. The latter lipopeptide, **C_12_-cTurg-1,** was the most potent in this series with a two-fold increased antimicrobial activity against four out of six strains, compared to its linear analogue **C_12_-Turg-1**. However, as opposed to **C_12_-Turg-1**, the cyclic analogue **C_12_-cTurg-1** displayed detectable haemolytic activity (EC_50_: 219 µg/mL).

Similar to the linear lipopeptides, we observed a small reduction in antimicrobial activity following the increase in the acyl chain length for the cyclic analogues **C_8_-cTurg-2**, **C_10_-cTurg-2** and **C_12_-cTurg-2**. Importantly, the cyclic lipopeptides in this series were more potent than their linear analogues. Moreover, this series included the overall most potent peptide prepared in this study, **C_8_-cTurg-2,** with a MIC of 2–4 µg/mL against all Gram-positive bacterial test strains and *E. coli*, and a MIC of 8 µg/mL against *P. aeruginosa*. Cys-Cys sidechain cyclization had a positive effect on the overall antimicrobial activity, most likely due to the formation of a more rigid cyclic structure. Somewhat surprisingly, **C_8_-cTurg-2** was considerably less haemolytic (EC_50_: 439 µg/mL) than the linear precursor lipopeptides **C_8_-, C_10_- and C_12_-Turg-2** (EC_50_: 55–198 µg/mL)**.**

Cyclic lipopeptides **C_8_-cTurg-6**, **C_10_-cTurg-6** and **C_12_-cTurg-6** were more potent than their linear analogues, but we also noticed a reduction in antimicrobial activity following the increase in the acyl chain length. Moreover, we observed an undesirable increase in haemolytic activity for this series of cyclic lipopeptides (EC_50_: 9–30 µg/mL), which, in contrast to antimicrobial activity, increased following acyl chain elongation. These results clearly demonstrate that optimization of the peptide’s activity involves a trade-off between achieving desired antimicrobial potency and minimizing unwanted toxicity against human RBCs.

### 2.5. SAR Summary

Our first series of cyclic peptides **cTurg-1**–**cTurg-7** demonstrated that substitution of Lys to Arg results in peptides with higher antimicrobial activity (Table 1). The cyclic Arg-modified peptides **cTurg-5**–**cTurg-7** were generally more potent than their Lys-containing counterparts **cTurg-2**–**cTurg-4**. The awareness of changes in haemolytic activity following Lys to Arg substitution is important, as shown for the Arg-modified cyclic peptide **cTurg-7** displaying weak haemolytic activity (EC_50_: 197 µg/mL), whereas its Lys analogue, **cTurg-4,** as well as other Trp and Arg modified cyclic peptides were non-haemolytic (EC_50_: ≥ 849 µg/mL).

For the linear lipopeptides **C_8_-**, **C_10_-** and **C_12_-Turg-1**, as well as the cyclic lipopeptides **C_8_-**, **C_10_-** and **C_12_-cTurg-1**, an increase in the number of carbons in the acyl chain resulted in increased potency of the corresponding analogues. A reverse trend was observed for both linear **C_8_-**, **C_10_-**, **C_12_-Turg-2** and cyclic **C_8_-**, **C_10_-**, **C_12_-cTurg-2** lipopeptides, as chain elongation beyond C_8_ resulted in analogues with mostly unchanged potency (C_10_-analogues), or even a two- to four-fold decrease in potency (C_12_-analogues). A similar observation was made for the linear **C_8_-**, **C_10_-**, **C_12_-Turg-6** and cyclic **C_8_-**, **C_10_-**, **C_12_-cTurg-6** lipopeptides, where the greatest decrease in potency was observed against *E. coli* (MIC: from 8 µg/mL to 64 µg/mL). The sequences of the linear and cyclic **C_8_-**, **C_10_-**, **C_12_-Turg-2/c-Turg-2** and **C_8_-**, **C_10_-**, **C_12_-Turg-6/c-Turg-6** lipopeptides were more hydrophobic than that of **C_8_-**, **C_10_-**, **C_12_-Turg-1/c-Turg-1**, due to their tryptophan-rich core sequence, which could, in part, explain the observed trend. As for the **C_8_-**, **C_10_-**, **C_12_- Turg-1/c-Turg-1** lipopeptides, it remains unclear whether the C_12_-chain conferred the threshold hydrophobicity, or whether this threshold could have been achieved by acylation with fatty acids containing more than 12 carbons. These results suggest that there might be an upper limit regarding hydrophobicity (threshold hydrophobicity) and that its further increase could have an unfavourable effect on antimicrobial activity, and in some cases even abrogate it entirely. This may occur, as proposed in a study done by Chu-Kung et al., when the minimal bactericidal concentration of the peptide is higher than its critical miscelle concentration [18]. Furthermore, studies have demonstrated that lipopeptides containing long fatty acid chains tend to self-assemble, resulting in reduced antimicrobial activity [19,20]. Our findings give further credence to the assumption that hydrophobicity, easily tuned by lipidation, is an important factor influencing antimicrobial activity of peptides.

The direct correlation between haemolytic activity and hydrophobicity (mirrored in the acyl chain length) of the peptides was especially prominent for the linear **C_8_-**, **C_10_-**, **C_12_-Turg-2** and **C_8_-**, **C_10_-Turg 6** lipopeptides, except for **C_12_-Turg-6,** which had a slightly reduced haemolytic activity than its analogue with **C_10_** fatty acid chain (Table 1). This trend was not observed for **C_8_-, C_10_-Turg-1**, and its cyclic analogues, as they were all non-haemolytic. In general, cyclic lipopeptides were more haemolytic compared to their linear analogues. These results support previous findings that lipopeptides with longer acyl chains have higher haemolytic activity, most likely due to their lower membrane selectivity [21,22].

While the effect of lipidation is shown to be bidirectional, depending on the initial hydrophobicity of the peptides, among other things, intramolecular cyclization led to increased antimicrobial activity of linear lipopeptides regardless of their primary sequence. For all cyclic lipopeptides synthesized in this work, the potency was either unchanged or improved four-fold, compared to their corresponding linear analogues. It should be noted that changes in the position of the acylation may also have bearing on the potency of lipopeptides, as well as head-to-tail cyclization, two strategies that remain to be explored. Our results are in line with previous research showing that acylation and intramolecular cyclization are useful tools important for fine-tuning antimicrobial and haemolytic activity of AMPs [17].

### 2.6. Antifungal Activity

The synthesised peptides were screened for antifungal activity against the molds *Aurobasidium pollulans* and *Rhodotorula* sp., and the yeast *Candida albicans*. All cyclic peptides of the **cTurg-1**–**cTurg-7** series displayed almost equal antifungal activity (MIC: 32–128 µg/mL) against all three fungi tested (Table 1). Even **cTurg-1,** containing the original Turgencin A core sequence, displayed antifungal activity against all strains at concentrations below the MIC against bacteria (except for *C. glutamicum*). None of the peptides of this series stood out as more potent than the others, indicating that the amino acid substitutions in the central core sequence are not important for antifungal activity.

All linear and cyclic lipopeptides prepared in this study displayed antifungal activity. In general, their MIC values ranged from 32 to 128 µg/mL against *A. pullulans* and *C. albicans*, making it difficult to conclude any structure–activity relationship for the peptides against these strains. The activity against *Rhodothorula* sp. varied more between the different peptides with MIC values from 4 to >128 µg/mL. The cyclic lipopeptides **C_8_-**, **C_10_-** and **C_12_-cTurg-1** were somewhat more potent against *Rhodothorula* sp. compared to the linear lipopeptides **C_8_-**, **C_10_-** and **C_12_-Turg-1**, while the longer C_10_/C_12_ analogues (MIC: 4 µg/mL) were slightly more potent compared to the corresponding C_8_ analogues. The linear lipopeptides **C_8_-**, **C_10_-** and **C_12_-Turg-2** were equally potent (MIC: 32 µg/mL) against *Rhodothorula* sp. However, among the cyclic lipopeptides, **C_8_-cTurg-2** was the most potent peptide (MIC: 4 µg/mL), followed by **C_10_-cTurg-2** (MIC: 16 µg/mL) and **C_12_-cTurg-2** (MIC: 32 µg/mL). These results support the antibacterial data, indicating an upper limit regarding lipophilicity. Among all lipopeptides tested, **C_8_-cTurg-2** was the most potent peptide against *A. pullulans* (MIC: 16 µg/mL). The linear lipopeptides **C_8_-**, **C_10_-** and **C_12_-Turg-6** were overall the least potent with MIC-values of 64 to >128 µg/mL against all test strains, but with a two- to four-fold increase in potency for their cyclic versions (**C_8_-**, **C_10_-** and **C_12_-cTurg-6**) against *A. pullulans* and *Rhodothorula* sp.

### 2.7. Selectivity Index

The selectivity index (SI) of the peptides towards bacteria over eukaryotic cells was calculated using the geometric mean (GM) of the MIC values against all bacterial test strains, according to the method by Orlov et al. [23]. The SI for each peptide was determined as the ratio of the RBC-EC_50_ value by the corresponding GM value. Larger SI values indicate greater selectivity for microbial cells [24]. As shown in Table 1, **cTurg-6** had the highest selectivity of the cyclic peptides (**cTurg-1** to **cTurg-7**), with an SI value of 138 for all bacterial strains tested. Interestingly, two of the lipopeptides, **C_10_-cTurg-1 and C_12_-Turg-1,** emerged as a promising candidates for further optimization, both having SI values for bacteria above 150. In addition, **C_8_-cTurg-2** showed the best selectivity profile among both cyclic and linear lipopeptides derived from the **cTurg-2** series. In general, the SI was higher against the Gram-positive bacteral strains than the Gram-negative strains. (Appendix A). Overall, peptides displaying EC_50_ < 100 µg/mL were considered too haemolytic to be of interest for further exploration.

### 2.8. Effects on Bacterial Viability and Membrane Integrity

We used two luciferase-based assays to investigate whether the synthesized peptides had an immediate effect on bacterial viability and membrane integrity. Changes in light emission of sensor bacteria constitutively expressing the bacterial lux operon or a eukaryotic luciferase can be used as a proxy for viability and membrane integrity, respectively [25]. Light production of the viability biosensors represents metabolic activity of the bacteria. For the membrane integrity assay on the other hand, light production depends on the influx of the externally added D-luciferin, which at neural pH will not readily pass the intact plasma membrane. An initial increase in light production therefore corresponds with damage to the plasma membrane and a concomitant influx of D-luciferin, while a subsequent drop in light emission indicates diminishing ATP reserves of the dying sensor bacteria.

Here we present the results for the two most potent cyclic lipopeptides, **C_12_-cTurg-1** and **C_8_-cTurg-2,** as well as for the membrane active agent chlorhexidine (CHX) that was used for comparison (Figure 3 and Figure 4). The results for the remaining peptides can be found in the Appendix A. The overall results for **C_12_-cTurg-1** and **C_8_-cTurg-2** in *B. subtilis* 168 show that increasing concentrations resulted in a decrease in light emisssion (and in increasing rates), suggesting a dose-dependent effect on viability (Figure 3).

In order to confirm that the observed decrease in viability was the result of membrane damage, we used the membrane integrity assay [13,25]. The results of the membrane integrity assay for *B. subtilis* 168 show that increasing lipophilicity of the cyclic lipopeptides **C_8_-, C_10_-,** and **C_12_-cTurg-1** caused increased membranolytic activity. A rapid and strong membrane disruptive effect was observed for the highly potent **C_12_-cTurg-1** peptide (Figure 3 and Appendix A). When analysing the membrane integrity effects of the cyclic lipopeptides **C_8_-, C_10_-** and **C_12_-cTurg-2**, increasing lipophilicity was also concordant with increased membrane activity, but not to a greater extent than for the **C_8_-, C_10_-** and **C_12_-cTurg-1** peptides. We observed a minor effect on increased lipophilicity on the membrane activity of **C_8_-, C_10_-** and **C_12_-cTurg-2** and **C_8_-, C_10_-** and **C_12_-cTurg-6** cyclic lipopetides, as they showed somewhat similar activity, showing a rapid decrease in light production, from concentrations 50 to 12.5 µg/mL (Figure 3 and Appendix A). At the lowest concentration, close to the MIC value of 2 µg/mL for all tested peptides, we observed minor changes in membrane activity and viability, most likely due to the high concentration of bacterial inoculum (1000-fold greater than that used for the MIC assay). This could explain why higher concentrations of the peptides were needed to see a more pronounced effect. However, at higher concentrations, the membranolytic action for some lipopeptides was so rapid (< 3 s) that the luminescence peak could not be detected, as the signal started declining even before the first measurement was made. This phenomenon was observed for **C_12_-cTurg-1** and **C_8_-cTurg-2** (at the highest test concentrations of 25 µg/mL and 50 µg/mL), indicating a more rapid or even different mechanism for disruption of membrane integrity than that of chlorhexidine (Figure 3).

The viability and membrane integrity assay results for *E. coli* K12 were quite different from what was observed for *B. subtilis*. For both **C_12_-cTurg-1** and **C_8_-cTurg-2,** we observed a gradual, dose-dependent reduction in viability in *E. coli*, although not as prominent as for *B. subtilis*. In the membrane integrity assay, **C_12_-cTurg-1** showed a delayed, 3.5-fold rise in luminescence at the highest test concentration (50 µg/mL) compared to chlorhexidine, with a subsequent decline in luminescence during the assay timeframe (Figure 4). A further delayed response was observed for **C_12_-cTurg-1** at 25 µg/mL, whereas no effect was observed at lower concentrations. **C_8_-cTurg-2** gave an even further delayed rise in peak luminescence in the membrane integrity assay, but only at the highest test concentration of 50 µg/mL (Figure 4). Although **C_12_-cTurg-1** and **C_8_-cTurg-2** displayed similar MIC values of 4 µg/mL against *E. coli* (ATCC 25922) in the screening assay run for 24 h, the results from the membrane integrity assay with *E. coli* K12 indicate a different mode of membrane disruption. This might suggest that the peptides were acting on the outer LPS and inner cytoplasmic membranes of *E. coli* at different rates, resulting in their delayed action observed in both the viability and membrane integrity assays.

### 2.9. Effects on E. coli Mutant Strain with an Impaired Outer Membrane

Antimicrobial activity of all synthesised cyclic peptides were determined against two additional *E. coli* strains: the hyperpermeable mutant strain NR698, and its isogenic wild type (WT) MC4100 (Table 2). The outer membrane deficiency of the mutant strain *E. coli* NR698 is based on the allele *imp*4213/*lptD*4213 constituting an in-frame deletion of the *imp* (increased membrane permeability) gene in *E. coli* [26]. It has been shown that *imp* mutations make the outer membrane more permeable to antibiotics like vancomycin, which normally does not readily cross the outer membrane barrier of *E. coli*. In addition, this mutation is also suggested to cause defects in LPS assembly [27,28]. The results from the previous screening against the laboratory strain *E. coli* ATCC 25922 are included in Table 2 for comparison and show that both the WT and mutant NR698 strains were in many cases more sensitive to the cyclic peptides than the *E. coli* ATCC 25922 laboratory strain.

The cyclic peptide **cTurg-1,** containing the native Turgencin A core sequence, was found to be active against the mutant NR698 strain (MIC: 64 µg/mL). However, against the WT strain and the *E. coli* ATCC 25922 laboratory strain it showed no activity (Table 2). In contrast to **cTurg-1,** Trp-modified cyclic peptides **cTurg-2** to **cTurg-4** displayed increased antimicrobial activity against all three *E. coli* strains, among which the mutant NR698 was most sensitive. Thus, for these peptides, the outer membrane appeared to hinder their antimicrobial effect. For the analogous, Arg-modified cyclic peptides **cTurg-5** to **cTurg-7**, no major differences in antimicrobial activity were observed against the three *E. coli* strains (MIC: 8–16 µg/mL), making these peptides seemingly less affected by the outer LPS membrane.

With regard to the cyclic lipopeptides (Table 2), the potency against the mutant strain NR698 showed the same trend as previously described (SAR section), although with lower MIC values. In brief, according to SAR, improved potency was achieved by increasing the acyl chain length for the cyclic lipopeptides **C_8_-**, **C_10_-** and **C_12_-cTurg-1**, while the opposite trend was observed for **C_8_-**, **C_10_-** and **C_12_-cTurg-2** and **C_8_-**, **C_10_-** and **C_12_-cTurg-6**. The results for the WT MC4100 strain are also in accordance with our previous SAR observations, but to a lesser degree than for the other two strains. These results clearly demonstrate that structural modifications can optimize target interactions and antibacterial potency as seen for example in **C_12_-cTurg-1, which** displays similar high potency against all three strains (MIC: 2–4 µg/mL).

In order to more closely evaluate the effect of an outer membrane on antimicrobial activity of synthesised peptides, we tested several commercially available antibiotics. Major improvement in antimicrobial activity was achieved against the mutant NR698 strain compared to the WT strain when treated with polymyxin B, vancomycin, ampicillin and chloramphenicol (Table 2). Compared to these antibiotics, several of the present cyclic peptides, such as **C_12_-cTurg-1** and **C_8_-cTurg-2**, showed similar or higher antibacterial activity against the *E. coli* ATCC 25922 and WT strains. In summary, the overall higher antibacterial activity against the mutant NR698 strain supports the hypothesis that the outer LPS membrane present in the WT strains could act as a barrier, limiting the effect of the synthesised peptides. This, in turn, may have affected the rate of bacterial membrane disruption as observed in the viability and membrane integrity studies.

### 2.10. Permeabilization of the Outer Membrane of E. coli

The outer membrane of Gram-negative bacteria acts as a barrier for many hydrophobic and larger hydrophilic substances (>600 Da) [29]. However, some peptides can sensitize the outer membrane and thus facilitate the entry of various hydrophobic molecules. To explore if the peptides affect the outer membrane of *E. coli* MC4100, **C_12_-cTurg-1** and **C_8_-cTurg-2** were tested for their ability to enable the entry of the hydrophobic fluorescent probe 1-*N*-phenylnapthylamine (NPN, MW of 219 Da) (Figure 5). In aqueous solutions, NPN shows very low fluorescence, which greatly increases upon interaction with the hydrophobic environment of biological membranes. Normally, the hydrophobic NPN is excluded by *E. coli* bacteria, but it can enter the bacteria once the integrity of the outer membrane is compromised. In this assay, both cyclic lipopeptides, **C_12_-cTurg-1** and **C_8_-cTurg-2**, as well as chlorhexidine, were found to increase the NPN fluorescence in a concentration-dependent manner (Figure 5), but at a slightly different rate, via membrane permeabilization. The stronger effect for **C_12_-cTurg-1**, with almost a four-fold increase in both fluorescence and luminescence for the concentrations of 25 to 50 µg/mL, suggests that **C_12_-cTurg-1** disrupts both the outer and the inner membrane at a similar rate at higher concentrations (6.3–12.5 × MIC). In contrast, **C_8_-cTurg-2** was found to alter the outer membrane permeability at concentrations that did not initially give any increase in luminescence in the inner membrane integrity assay (Figure 4 and Figure 5). Thus, the observed effects indicate that the outer membrane passage of **C_8_-cTurg-2** was a rate-limiting step that was most likely preventing the peptide from reaching and accumulating in the inner membrane.

### 2.11. Bacterial Killing Experiments

The two most potent lipopeptides (**C_12_-cTurg-1** and **C_8_-cTurg-2**) were selected for bacterial killing experiments to see whether the peptides displayed bacteriostatic or bactericidal effects on the bacterial inoculum used in the MIC assay. In this experiment, 10 µL aliquots from the wells containing peptide-treated bacteria were harvested after the MIC assay (24 h incubation), and 10-fold serially diluted and spotted on MH agar plates for colony counting. At their half-MIC concentration, slightly less colony-forming units (CFU) were formed for both *S. aureus* (ATCC 9144) and *E. coli* (ATCC 25922) in the presence of **C_12_-cTurg-1** and **C_8_-cTurg-2**, compared to the growth control (Figure 6). No CFU were observed at their MIC (or concentrations above MIC). These results show that the peptides displayed a bactericidal action on the bacterial strains tested.

## 3. Materials and Methods

### 3.1. Peptide Synthesis

All reagents and solvents were purchased from commercial sources and used as supplied. All peptides were synthesized using standard Fmoc-solid phase methodology using a Rink Amide ChemMatrix resin (loading 0.50 mmol/g). The resin was pre-swelled in *N,N*-dimethylformamide (DMF, 4.5 mL) for 20 min at 70 °C. The Fmoc-protected amino acids (4.0 eq.), saturated fatty acids (4.0 eq.) and O-(6-chlorobenzotriazol-1-yl)*N,N,N’,N’*-tetramethyl-uroniumhexafluorophosphate (HCTU, 4 eq.) were dissolved in DMF to a concentration of 0.5 M, 0.5 M and 0.6 M, respectively. *N,N*-Diisopropylethylamine (DIEA, 8 eq.) was dissolved in *N*-methyl-2-pyrrolidone (NMP) to a concentration of 2.0 M. Coupling steps for all amino acids except cysteine were performed under microwave conditions at 75 °C, for 10 min. To avoid racemization of Cys and Arg side-reaction due to microwave heating, Fmoc-Cys(Trt)-OH and Fmoc-Arg(Pbf)-OH were coupled at r.t. for 60 min. For the N-terminally acylated lipopeptides, the coupling reaction with HCTU was performed twice to ensure successful attachment of the acyl chain to the peptides. Following each coupling step, the resin was washed 4 times with DMF (4.5 mL) for 45 s. After the desired linear peptide was assembled, the resin was washed first with dichloromethane (DCM, 4.5 mL) for 45 s (6 times), and then with diethyl ether (3–4 times). The resin was dried on a vacuum manifold and placed in a desiccator overnight.

### 3.2. Fmoc Deprotection

The peptides were deprotected and cleaved from the resin using a standard cleavage cocktail consisting of trifluoracetic acid (TFA), Milli-Q ultrapure water, 1,2-ethanedithiol (EDT) and triisopropylsilane (TIS) (TFA/water/EDT/TIS; 94/2.5/2.5/1.0 (*v/v/v/v*) at room temperature. The cleavage procedure was repeated twice, each time with 10 mL of the cleavage mixture under occasional stirring. Following the first 3 h cleavage step, the resin was collapsed with a small amount of DCM, and then washed with diethyl ether. The second cleavage step was performed with the same amount of cleavage cocktail (10 mL) for an additional 1 h. After each cleavage step, as well as after addition of DCM and diethyl ether, the resin was dried under a vacuum. The collected filtrates containing the desired peptide were pooled into a round bottom flask and the solvents were evaporated, resulting in a thin, glassy film covering the walls of the flask. Precipitation of the peptides was induced by the addition of ice-cold diethyl ether, which was decanted after 24 h. This procedure was repeated twice, and the residual diethyl ether was evaporated before purification.

### 3.3. Peptide Purification by Preparative Reversed-Phase High-Performance Liquid Chromatography (RP-HPLC)

Crude peptide purification was performed by RP-HPLC using a preparative SunFire C_18_ OBD, 5 μm, 19 × 250 mm column (Waters, Milford, MA, USA at room temperature. The HPLC system (Waters) was equipped with a 2702 autosampler, a 2998 photodiode array (PDA) detector and an automated fraction collector. The lipopeptides were purified using a linear gradient of eluent A (water with 0.1% TFA) and eluent B (acetonitrile with 0.1% TFA), ranging from 20–60% B, over 25 min, at a flow rate of 10 mL/min. Purified fractions were collected and freeze-dried prior to further characterization.

### 3.4. Peptide Characterization by High-Resolution Mass Spectrometry (HRMS)

The purified peptides were characterized by HRMS using an Orbitrap Id-X Tribrid mass analyser equipped with an electrospray ionization (ESI) source (Thermo Fischer Scientific, Waltham, MA, USA), with a Vanquish UHPLC system (Waters), coupled to an Acquity Premier BEH C_18_, 1.7 μm, 2.1 × 100 mm column (Waters). The ESI mass spectra were obtained in positive ion mode. Prior to analysis, all samples were dissolved in 1 mL of Milli-Q water. The lipopeptides were eluted with a 0.5–95.0% linear gradient of the eluent B (A: water with 0.1% formic acid, B: acetonitrile with 0.1% formic acid) over 10 min, with a flow rate of 0.5 mL/min. The injection volume was 2 μL, and the column temperature was set to 60 °C.

### 3.5. Purity Determination by Ultra-Performance Liquid Chromatography (UPLC)

Purity of the synthetized peptides was determined using an analytical UPLC-PDA H-class system (Waters, Milford, MA, USA). The analysis was conducted on an Acquity UPLC BEH 1.7 μm, 2.1 ×100 mm C_18_ column, using a linear gradient of eluent A (water with 0.1% TFA) and eluent B (acetonitrile with 0.1% TFA), ranging from 0.5–95.0% B, over 10 min. The flow rate was set to 0.5 mL/min and the temperature of the column was set to 60 °C. A 2996 PDA detector with a wavelength ranging from 210–400 nm was used to record the UV absorbance of the purified peptides. Retention times for each peptide were recorded as a measurement of hydrophobicity.

### 3.6. Cys-Cys Cyclization

The cyclization process included intramolecular disulphide formation between the sulfhydryl (SH) side chains of two cysteine residues, Cys17 and Cys26. This step was performed on a parallel reaction station, either under open air or with a continuous supply of oxygen by careful bubbling. The lipopeptides (5 mg) were dissolved in Milli-Q water to a concentration of 250 μg/mL. The reaction proceeded at room temperature under continuous magnetic stirring. The progress of the reaction was monitored by UPLC-HRMS. Upon completion of the reaction, peptide solutions were lyophilized, and their purity determined as described above.

### 3.7. Antibacterial Minimum Inhibitory Concentration (MIC) Assay and Killing Assay

All cyclic and linear peptides were screened for antibacterial activity against four Gram-positive strains: *B. subtilis* (Bs, ATCC 23857), *C. glutamicum* (Cg, ATCC 13032), *S. aureus* (Sa, ATCC 9144) and *S. epidermidis* RP62A (Se, ATCC 35984), and two Gram-negative strains: *E. coli* (Ec, ATCC 25922) and *P. aeruginosa* (Pa, ATCC 27853). The activity was assessed using a broth microdilution assay according to a modified CLSI-based method [30]. Briefly, overnight bacterial cultures were grown in Mueller–Hinton (MH) broth medium (Difco Laboratories, Detroit, MI, USA) for 2 h at room temperature. The optical density (OD_600_) was measured, and the bacterial suspensions were adjusted to 2.5–3 × 10^4^ CFU/mL in MH medium. The bacterial suspension (50 µL) was distributed in 96-well plates (Nunc, Roskilde, Denmark) preloaded with two-fold dilution series (256 to 1 µg/mL) of peptide solutions (50 µL), giving a final well volume of 100 µL. The microplates were incubated in an EnVision 2103 microplate reader (PerkinElmer, Llantrisant, UK) at 35 °C, with OD_595_ recorded every hour for 24 h. MIC was defined as the lowest concentration of peptides showing an optical density less than 10% of the negative (growth) control, consisting of bacterial suspension and water. Polymyxin B sulfate (Sigma-Aldrich, St. Louis, MO, USA) and chlorhexidine acetate (CHX, Fresenius Kabi, Halden, Norway), both with concentrations ranging from 12.5 to 0.09 μg/mL, served as positive (growth inhibition) controls. All peptides were tested in triplicate.

A killing experiment was performed on selected lipopeptides by using actively growing cultures of *S. aureus* (ATCC 9144) and *E. coli* (ATCC 25922). The procedure was performed as previously described [31]. Briefly, after 24 h of peptide treatment (MIC assay, as described above), aliquots (10 μL) of 10-fold serial dilutions (in MH broth) of wells containing ½, 1 and 2 × MIC of the peptides (with bacteria) were plated on MH Agar (Difco) plates. The number of colony-forming units (CFU) was determined after 24 h of incubation at 37 °C. The tests were performed in triplicate.

### 3.8. Antifungal MIC Assay

The synthesized peptides were screened for antifungal activity against the molds *A. pullulans* (Ap) and *Rhodotorula* sp. (Rh) (both obtained from Professor Arne Tronsmo, The Norwegian University of Life Sciences, Ås, Norway) and the yeast *C. albicans* (Ca, ATCC 10231) as previously described [13]. In short, fungal spores were grown in potato dextrose broth media (Difco) containing 2% D(+)-glucose (Merck, Darmstadt, Germany) at 25–30 °C while shaking at 200 rpm overnight. The cultures were diluted with a dextrose media containing glucose to a concentration of approx. 4 × 10^5^ spores/mL. Aliquots of the cultures (50 µL) were transferred to 96 well microtiter plates preloaded with the synthetic peptides (50 µL) in two-fold serial dilutions (256 to 1 µg/mL). Polymyxin B and CHX (both with concentrations ranging from 12.5 to 0.09 μg/mL) served as positive antibiotic controls. The microtiter plates containing the fungal spores and the test peptides were incubated at room temperature for 48 h and OD_600_ was recorded using a Synergy H1 Hybrid microplate reader system (BioTek, Winooski, VT, USA). MIC was defined as the lowest peptide concentration showing an optical density less than 10% of the negative (growth) control. All experiments were performed in triplicate.

### 3.9. Haemolytic Activity Assay

The synthetized lipopeptides were screened for haemolytic activity against human red blood cells (RBC), in concentrations ranging from 500 to 3.9 μM, following a previously described protocol [32]. In brief, haemolysis was determined using a heparinizied (10 IU/mL) fraction of freshly drawn human blood. A second fraction of blood, which was collected in test tubes containing ethylenediaminetetraacetic acid (EDTA, Vacutest, KIMA, Arzergrande, Italy), was used for determination of the hematocrit (hct). Plasma was removed from heparinized blood by washing three times with prewarmed phosphate-buffered saline (PBS) before being adjusted to a hematocrit of 4%. Peptides dissolved in dimethyl sulfoxide (DMSO) were further diluted with PBS to a final DMSO content of ≤1%. Triton X-100 (Sigma-Aldrich, St. Louis, MO, USA), used at a final concentration of 0.1%, served as a positive control for 100% haemolysis, whereas 1% DMSO in PBS buffer served as a negative control where no toxicity was detected. Duplicates of test solutions and erythrocytes, with 1% hct final concentration, were prepared in a 96-well polypropylene V-bottom plate (Nunc, Fischer scientific, Oslo, Norway). They were incubated under agitation at 37 °C and 800 rpm for 1 h. After centrifugation (5 min, 3000× *g*), 100 µL from each well were transferred to a flat-bottomed 96-well plate and absorbance was measured at 545 nm with a microplate reader (SpectraMax 190, Molecular Devices, San Jose, CA, USA). After subtracting PBS background, the percentage of haemolysis was calculated as the ratio of the absorbance in the peptide-treated and surfactant-treated samples. Three independent experiments were performed, and EC_50_ (the concentration giving 50% haemolysis) values are presented as averages.

### 3.10. Bacterial Biosensor Membrane Integrity Assay

A membrane integrity assay was performed using two bacterial biosensors, *B. subtilis* 168 (ATCC 23857) and *E. coli* K12 (ATCC MC1061). Both strains carry the pCSS962 plasmid that contains a eukaryotic luciferase gene *(lucGR*), which originate from the click beetle *Pyrophorus plagiophthalamus* [33]. The assay was performed as described previously [13]. Briefly, overnight cultures grown in MH media in the presence of respective antibiotics such as 5 µg/mL chloramphenicol (Merck, Darmstadt, Germany) for *B. subtilis* and both 20 µg/mL chloramphenicol and 100 µg/mL ampicillin (Sigma-Aldrich) for *E. coli*, were further diluted in fresh MH media without antibiotics and grown until they reached an OD_600_ of 0.1. D-luciferin potassium salt (Synchem Inc., Elk Grove Village, IL, USA) was added to the cell suspension at a final concentration of 1 mM. Two-fold dilutions (final assay concentration of 50–1.56 µg/mL) of peptides dissolved in Milli-Q water were prepared and added (10 µL per well) to black round-bottom 96-well microtiter plates (Nunc). Milli-Q water served as a negative control used for the normalization purpose and CHX, having known membranolytic activity, was used as a positive control. The plates were loaded into a Synergy H1 Hybrid Reader and the background luminescence was monitored before aliquots (90 µL) of the cell suspension with D-luciferin were added, one well at the time, by an automatic injector. Light emission was recorded every second for 3 min. Each study was performed at least three times independently, and the figures show a representative dataset.

### 3.11. Bacterial Biosensor Viability Assay

Bacterial viability, based on light production by constitutively expressed bacterial luciferase, was measured in real time according to the method described by Hansen et al. [13]. The assay was performed using *B. subtilis* 168 with chromosomally integrated *lux*ABCDE operon and *E. coli* K12 carrying the plasmid pCGLS-11 containing the *lux*CDABE operon. The assay set up was the same as for the membrane integrity assay, with the exception that *B. subtilis* 168 and *E. coli* colonies were grown in MH media supplemented with 5 µg/mL chloramphenicol and 100 µg/mL ampicillin, respectively. No D-luciferin was added to the cell suspension. Each assay was performed at least three times independently, and the figures show a representative dataset.

### 3.12. Screening for Activity against E. coli Mutants

MIC values were determined for all cyclic peptides against two *E. coli* strains: wild type (WT) MC4100 and the hyperpermeable variant NR698, having a deficient outer membrane. The NR698 strain, containing *lptD*4213/*imp*4213 mutation, was kindly provided by M. Grabowicz (Emory University School of Medicine, Rollins Research Center, Atlanta, GA, USA) [28]. The assay was performed in the same way as previously described for the antibacterial MIC assay. Vancomycin hydrochloride (Hospira Enterprises BV, Almere, The Netherlands) and ampicillin: inhibitors of peptidoglycan synthesis, chloramphenicol: inhibitor of protein synthesis, and CHX and polymyxin B: membrane active compounds, were used as reference antibiotics to evaluate the permeability defects in *E. coli* NR698.

### 3.13. Outer Membrane Permeability Assay

The permeability of the *E. coli* outer membrane was analysed by measuring increased fluorescence as kinetics of 1-N-phenylnaphthylamine (NPN, Sigma-Aldrich) uptake following the protocol described by [34], with minor modifications. Briefly, a single colony of *E. coli* MC4100 (WT) was suspended in MH medium and incubated overnight at 37 °C with shaking (225 rpm). The culture was diluted in MH medium and adjusted to OD_600_ = 0.1 and incubated at 37 °C until it reached an OD_600_ of 0.5. The cells were centrifuged and washed twice with an assay buffer (5 mM HEPES, 5 mM glucose, pH 7.2) and resuspended in the same buffer to a final OD_600_ of 1.0. *E. coli* MC4100 cells were mixed with 20 µM NPN. The assay set up was the same as for the membrane integrity and viability assay, using black round-bottom 96-well microtiter plates (Nunc) containing 10 µL of 2-fold dilutions of peptides (500 to 31.2 µg/mL). A volume of 90 µL of cell suspension with NPN was added to each well by the automated injector of the Synergy H1 Hybrid Reader. The fluorescence was immediately measured (well by well) at an excitation wavelength of 350 nm and an emission wavelength of 420 nm every second for 3 min. The relative fluorescence was calculated by normalizing the values from each time point with the negative control (Milli-Q water). CHX was included as a positive control.

## Figures and Tables

**Figure 1 ijms-23-13844-f001:**
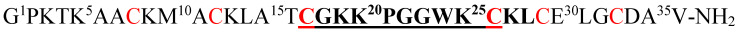
Amino acid sequence of Turgencin A with the 12-residue loop region in bold (residues 17–28). Cys-Cys connectivity in the loop region is underlined. Disulphide connectivity in the native Turgencin A peptide is Cys8-Cys33, Cys12-Cys29 and Cys17-Cys26 [12].

**Figure 2 ijms-23-13844-f002:**
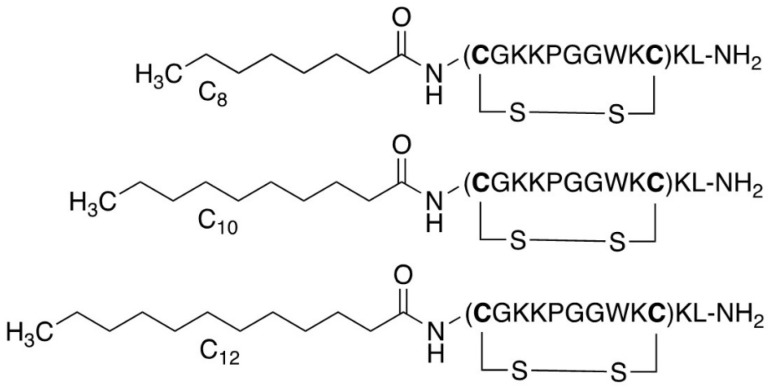
Lipopeptide modifications exemplified for **cTurg-1** lipopeptides containing C_8_, C_10_ and C_12_ fatty acids.

**Figure 3 ijms-23-13844-f003:**
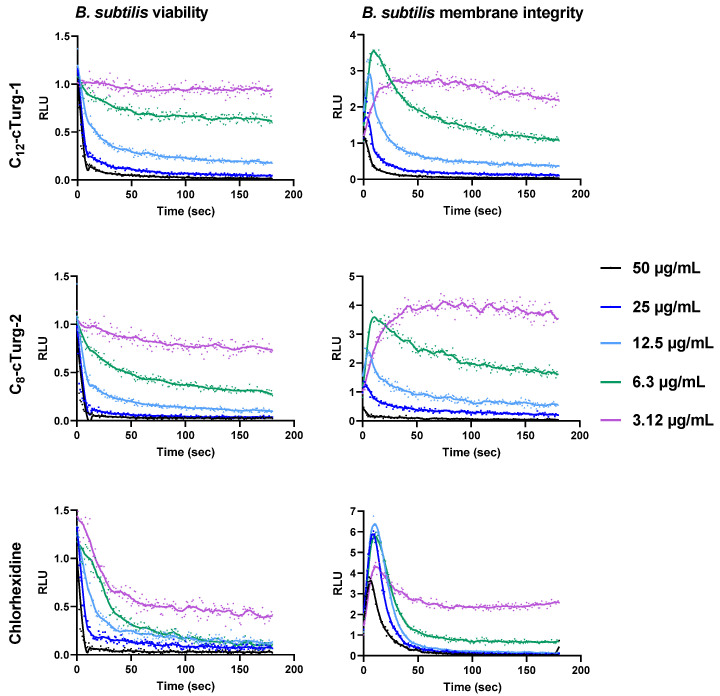
Effects of **C_12_-cTurg-1**, **C_8_-cTurg-2** and chlorhexidine on the kinetics of viability (**left**) and membrane integrity (**right**) in *B. subtilis* 168. Light emission normalized to the untreated water control is plotted as relative luminescence emission (RLU) over time (seconds). Kinetics of the immediate effect (within 3 min) on bacterial viability and membrane integrity, as measured by relative luminescence emission in *B. subtilis* 168 treated with increasing concentrations of the lipopeptides. Chlorhexidine served as a positive (membranolytic) control and water as a negative (untreated) control. All the graphs of this figure show a representative data set where each experiment was run at least three times independently.

**Figure 4 ijms-23-13844-f004:**
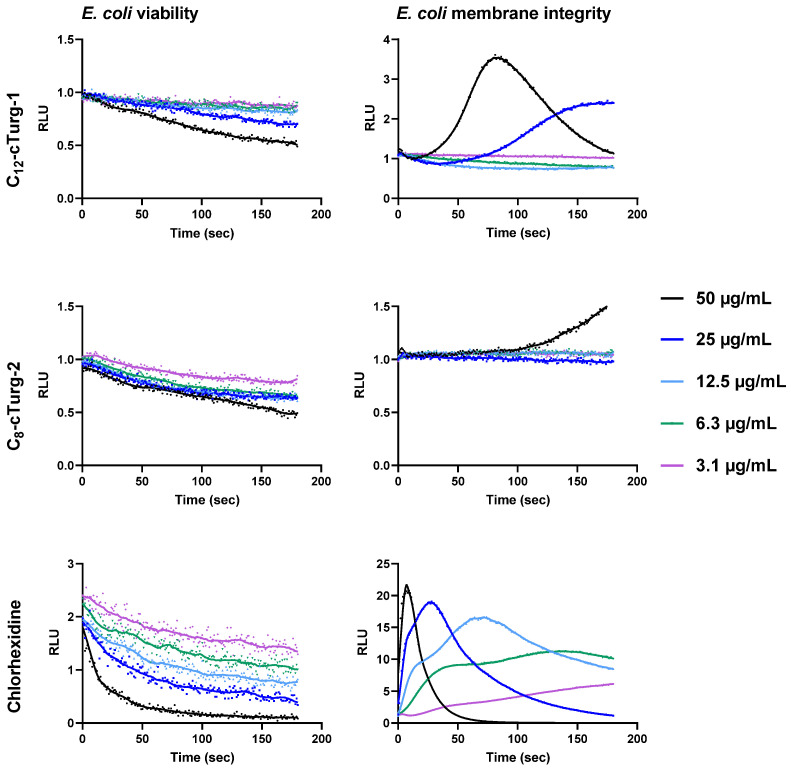
Effects of **C_12_-cTurg-1**, **C_8_-cTurg-2** and chlorhexidine on the kinetics of viability (**left**) and membrane integrity (**right**) in *E. coli* K12. Light emission normalized to the untreated water control is plotted as relative luminescence emission (RLU) over time (seconds). After addition of the bacterial inoculum (mixed with 1 mM D-luciferin in the membrane assay) to the wells, preloaded with lipopeptides, the light emission was measured each second for three min. Each colored line represents the total 180 s data points (mean of three independent measurements) from the assay at different concentration of the lipopeptides.

**Figure 5 ijms-23-13844-f005:**
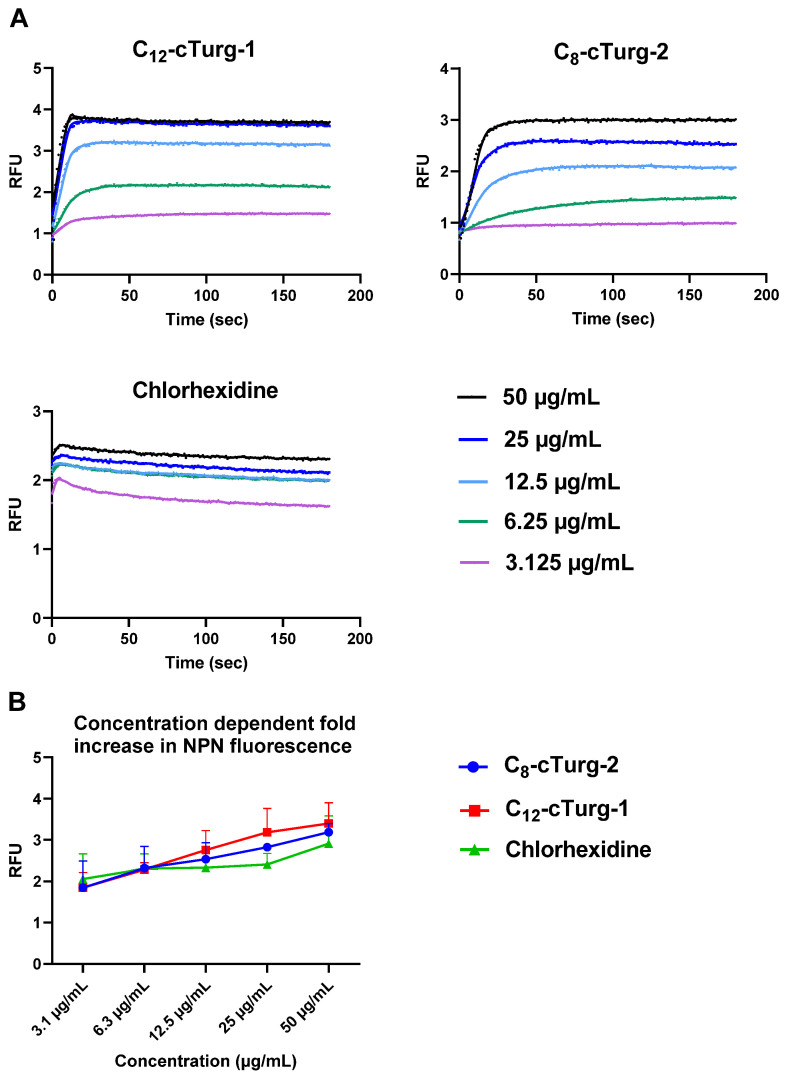
(**A**) Comparison of the effects of **C_12_-cTurg-1**, **C_8_-cTurg-2** and chlorhexidine on the kinetics of NPN fluorescence in *E. coli* MC4100 (WT). After addition of the bacterial inoculum (mixed with 20 µM NPN) to the wells (preloaded with lipopeptides), light emission was measured each second for 3 min. Each colored line represents the total 180 s data points from the assay at different concentrations. Each figure shows a representative data set. (**B**) *E. coli* MC4100 grown in MH media were treated with different concentrations of **C_12_-cTurg-1**, **C_8_-cTurg-2** and chlorhexidine. The permeability of the outer membrane was assessed by measuring the fluorescence of NPN after 3 min (mean of three independent measurements). In all data sets, fluorescence values were compared to bacterial cells treated with the same amount of Mill-Q water control.

**Figure 6 ijms-23-13844-f006:**
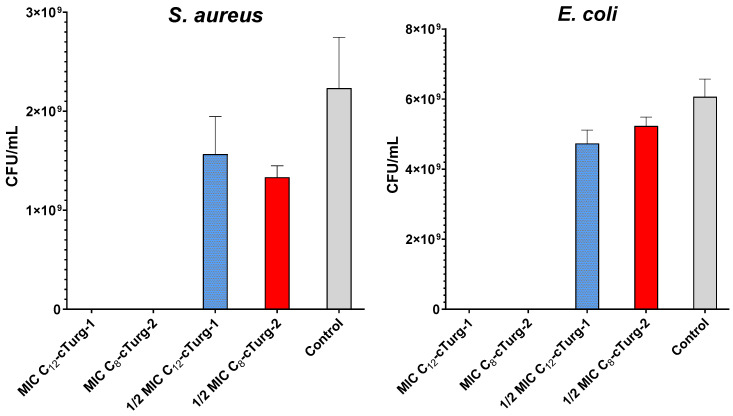
Bactericidal activity of **C_12_-cTurg-1** and **C_8_-cTurg-2** against *S. aureus* and *E. coli.* Colony-forming units (CFU) per mL bacterial inoculum were counted after treatment with MIC (4 µg/mL), 1/2 × MIC and no treatment (Control). Each bar presents the mean of three replicates ± SD.

**Table 1 ijms-23-13844-t001:** Antimicrobial activity (MIC in µg/mL), haemolytic activity against human RBC (EC_50_ in µg/mL) and selectivity index (SI). Sequence modifications (Trp and Arg replacements) compared to Turgencin A are shown in bold, and sequences in parentheses denote Cys-Cys cyclic peptides. SI was calculated as the ratio between haemolytic activity (EC_50_) and the geometric mean (GM) of the MIC values against all bacterial strains, i.e., SI = EC_50_/GM.

							Antimicrobial Activity (MIC) ^1^										RBC	
Peptide			Sequence	Mw ^2^	Net Charge ^3^	Rt ^4^	Gram +				Gram −		GM	Fungi			Tox. (EC_50_)	SI
							Bs	Cg	Sa	Se	Ec	Pa		Ap	Ca	Rh		
Cyclic peptides		cTurg-1	(CGKKPGGWKC)KL-NH_2_	1301.6	+5	3.11	256	16	>256	>256	>256	>256	161	32	128	64	nt ^5^	nt
	W	cTurg-2	(CGKK**WW**GWKC)KL-NH_2_	1519.9	+5	3.87	8	4	32	16	64	64	20	32	32	32	>1045	>52
		cTurg-3	(CGKK**W**G**W**WKC)KL-NH_2_	1519.9	+5	3.92	4	4	32	16	32	128	18	32	32	32	849	47
		cTurg-4	(CGKKP**WW**WKC)KL-NH_2_	1560.0	+5	3.97	8	4	64	32	64	256	32	32	64	32	>1065	>33
	R/W	cTurg-5	(CG**RRWW**GW**R**C)**R**L-NH_2_	1632.0	+5	3.98	8	4	16	8	8	16	9	32	32	32	>1101	>123
		cTurg-6	(CG**RRW**G**W**W**R**C)**R**L-NH_2_	1632.0	+5	4.02	4	4	16	8	8	16	8	32	32	32	1101	138
		cTurg-7	(CG**RR**P**WW**W**R**C)**R**L-NH_2_	1672.0	+5	4.09	4	4	16	8	16	32	10	32	32	32	197	20
Linear lipopeptides		C_8_-Turg-1	C_8_- CGKKPGGWKC KL-NH_2_	1429.8	+4	4.38	8	4	128	32	32	128	29	32	128	16	>943	>33
		C_10_-Turg-1	C_10_- CGKKPGGWKC KL-NH_2_	1457.9	+4	4.89	4	4	16	8	16	32	10	32	64	8	>957	>95
		C_12_-Turg-1	C_12_- CGKKPGGWKC KL-NH_2_	1486.0	+4	5.44	4	4	8	4	8	16	6	32	64	8	>971	>153
	W	C_8_-Turg-2	C_8_- CGKK**WW**GWKC KL-NH_2_	1648.1	+4	4.97	8	4	8	8	16	16	9	32	64	32	198	22
		C_10_-Turg-2	C_10_- CGKK**WW**GWKC KL-NH_2_	1676.2	+4	5.41	8	4	8	8	8	16	8	32	64	32	64	8
		C_12_-Turg-2	C_12_- CGKK**WW**GWKC KL-NH_2_	1704.2	+4	5.89	8	16	16	8	16	32	14	32	64	32	55	4
	R/W	C_8_-Turg-6	C_8_- CG**RRW**G**W**W**R**C **R**L-NH_2_	1760.2	+4	5.07	4	4	16	8	8	32	9	128	64	128	54	6
		C_10_-Turg-6	C_10_- CG**RRW**G**W**W**R**C **R**L-NH_2_	1788.2	+4	5.51	8	16	32	16	32	64	23	128	64	128	21	1
		C_12_-Turg-6	C_12_- CG**RRW**G**W**W**R**C **R**L-NH_2_	1816.3	+4	5.98	16	16	32	16	64	128	32	128	64	>128	39	1
Cyclic lipopeptides		C_8_-cTurg-1	C_8_ -(CGKKPGGWKC)KL-NH_2_	1427.8	+4	4.27	4	4	128	32	32	128	25	64	64	8	>942	>37
		C_10_-cTurg-1	C_10_ -(CGKKPGGWKC)KL-NH_2_	1455.9	+4	4.74	2	2	16	4	8	32	6	32	64	4	>956	>151
		C_12_-cTurg-1	C_12_ -(CGKKPGGWKC)KL-NH_2_	1483.9	+4	5.22	2	2	4	4	4	16	4	32	64	4	219	55
	W	C_8_-cTurg-2	C_8_- (CGKK**WW**GWKC)KL-NH_2_	1646.1	+4	4.70	2	2	4	4	4	8	4	16	32	4	439	123
		C_10_-cTurg-2	C_10_- (CGKK**WW**GWKC)KL-NH_2_	1674.1	+4	5.11	2	4	4	4	8	8	5	32	64	16	106	24
		C_12_-cTurg-2	C_12_- (CGKK**WW**GWKC)KL-NH_2_	1702.2	+4	5.55	4	8	8	8	16	16	9	32	64	32	32	4
	R/W	C_8_-cTurg-6	C_8_- (CG**RRW**G**W**W**R**C)**R**L-NH_2_	1758.2	+4	4.85	4	4	8	4	8	16	6	64	64	32	30	5
		C_10_-cTurg-6	C_10_- (CG**RRW**G**W**W**R**C)**R**L-NH_2_	1786.2	+4	5.28	4	4	8	8	16	16	8	64	64	32	16	2
		C_12_-cTurg-6	C_12_- (CG**RRW**G**W**W**R**C)**R**L-NH_2_	1814.3	+4	5.72	8	8	16	8	32	32	14	64	128	64	9	1
		Polymyxin B		1301.6	+5		3.1	3.1	12.5	6.3	3.1	3.1	161	3.1	12.5	3.1	nt	nt
		Chlorhexidine		505.5	+2		1.6	0.8	1.6	1.6	1.6	6.3	20	1.0	7.8	1.0	nt	nt

^1^ Microbial strains; Bs—*Bacillus subtilis*, Cg—*Corynebacterium glutamicum*, Sa—*Staphylococcus aureus*, Se—*Staphylococcus epidermidis*, Ec—*Escherichia coli*, Pa—*Pseudomonas aeruginosa*, Ap—*Aurobasidium pollulans*, Ca—*Candida albicans*, Rh—*Rhodotorula* sp. ^2^ Average molecular mass without including a TFA salt for each cationic charge. ^3^ Net charge at physiological pH (7.4). ^4^ Hydrophobicity measured as retention time (Rt; min) on a RP-UPLC C_18_ column using a linear acetonitrile/water gradient. ^5^ nt: not tested.

**Table 2 ijms-23-13844-t002:** Antimicrobial activity (MIC in µg/mL) of cyclic peptides and selected antibiotics against three strains of *E. coli* for investigation of effects concerning outer membrane permeability. In addition to the laboratory strain *E. coli* ATCC 25922, the activity against *E. coli* MC4100 (wild type, WT) and the outer membrane permeable mutant *E. coli* NR698 was measured and compared.

			*E. coli* Strains		
			ATCC 25922 (from Table 1)	MC4100 (WT)	NR698 ^1^ (mutant)
		Peptide			
Cyclic peptides		cTurg-1	>256	>256	64
	W	cTurg-2	64	16	8
		cTurg-3	32	16	8
		cTurg-4	64	16	8
	R/W	cTurg-5	8	8	8
		cTurg-6	8	8	8
		cTurg-7	16	8	8
Cyclic lipopeptides		C_8_-cTurg-1	32	16	8
		C_10_-cTurg-1	8	8	4
		C_12_-cTurg-1	4	4	2
	W	C_8_-cTurg-2	4	8	2
		C_10_-cTurg-2	8	8	4
		C_12_-cTurg-2	16	8	8
	R/W	C_8_-cTurg-6	8	8	8
		C_10_-cTurg-6	16	8	8
		C_12_-cTurg-6	32	16	16
Antibiotics		Polymyxin B	3.1	0.8	0.2
		Chlorhexidine	1.6	0.3	0.3
		Vancomycin	125	64	0.3
		Ampicillin	8	8	0.3
		Chloramphenicol	1.8	12.5	3.1

^1^*E. coli* MC4100 NR698 *imp*4213 (with deficient outer membrane).

## Data Availability

Not applicable.

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
