# Peer review of "Synthesis and Antimicrobial Activity of Short Analogues of the Marine Antimicrobial Peptide Turgencin A: Effects of SAR Optimizations, Cys-Cys Cyclization and Lipopeptide Modifications"

_ijms, 2022, doi:10.3390/ijms232213844_

Round 1

Reviewer 1 Report

Authors performed the peptide synthesis at 70 oC, is there any reason to perform at 70 oC instead of room temparature? (3.1)

Reviewer 2 Report

I am writing about the manuscript entitle: Synthesis and antimicrobial activity of short analogues of the marine antimicrobial peptide Turgencin A: Effects of SAR-optimizations, Cys-Cys cyclization and lipopeptide modifications. 

The use of antimicrobial peptides (AMPs) to counteract the Antimicrobial resistance (AMR) is very interesting. The authors have synthesised short analogues of the marine antimicrobial peptide Turgencin A from the colonial Arctic ascidian Synoicum turgens showing bactericidal effect and antifungal activity.  The article is well constructed, the experiments were well conducted, and analysis was well performed. The tables and the figures are clear and complete with data. The results have been discussed and explained in the discussion.

In my opinion the ms can be accepted in present form for publication, except for a few small tips:

Title

The title is too long, it could be shortened: “Synthesis and antimicrobial activity of short analogues of the marine antimicrobial peptide Turgencin A”.

Introduction

-Line 37: please correct the sentence (including fish, birds and mammals).

Results and Discussion

-Line 261 this sentence is repeated: we used two luciferase-based assays.